# New Electronic Tongue Sensor Array System for Accurate Liquor Beverage Classification

**DOI:** 10.3390/s23136178

**Published:** 2023-07-05

**Authors:** Jersson X. Leon-Medina, Maribel Anaya, Diego A. Tibaduiza

**Affiliations:** 1Department of Mechanical and Mechatronics Engineering, Universidad Nacional de Colombia-Sede Bogotá, Bogotá 111321, Colombia; jxleonm@unal.edu.co; 2Control, Data and Artificial Intelligence (CoDAlab), Department of Mathematics, Escola d’Enginyeria de Barcelona Est (EEBE), Universitat Politècnica de Catalunya (UPC), 08019 Barcelona, Spain; 3Department of Electrical and Electronic Engineering, Universidad Nacional de Colombia-Sede Bogotá, Bogotá 111321, Colombia; manaya@unal.edu.co

**Keywords:** sabajon, classification, machine learning, dimensionality reduction, manifold learning, electronic tongue

## Abstract

The use of sensors in different applications to improve the monitoring of a process and its variables is required as it enables information to be obtained directly from the process by ensuring its quality. This is now possible because of the advances in the fabrication of sensors and the development of equipment with a high processing capability. These elements enable the development of portable smart systems that can be used directly in the monitoring of the process and the testing of variables, which, in some cases, must evaluated by laboratory tests to ensure high-accuracy measurement results. One of these processes is taste recognition and, in general, the classification of liquids, where electronic tongues have presented some advantages compared with traditional monitoring because of the time reduction for the analysis, the possibility of online monitoring, and the use of strategies of artificial intelligence for the analysis of the data. However, although some methods and strategies have been developed, it is necessary to continue in the development of strategies that enable the results in the analysis of the data from electrochemical sensors to be improved. In this way, this paper explores the application of an electronic tongue system in the classification of liquor beverages, which was directly applied to an alcoholic beverage found in specific regions of Colombia. The system considers the use of eight commercial sensors and a data acquisition system with a machine-learning-based methodology developed for this aim. Results show the advantages of the system and its accuracy in the analysis and classification of this kind of alcoholic beverage.

## 1. Introduction

Recent advances in integrating sensors and data processing techniques have enabled the development of smart systems with the capability of analysis and data management [1] to make decisions and provide early alerts or bring relevant information about a specific process. These capabilities are ideal for process monitoring, condition monitoring, and abnormality detection, among others [2], because they can improve the confidence of the process and provide feedback on the current state of its variables [3]. To ensure the necessary quality of a process, monitoring involving the use of multiple sensors [2] with redundant information, sensor data fusion strategies, among other advanced strategies, can be utilized to avoid false alarms or information that can produce erroneous decisions for the control and quality tasks during production. One example of this kind of process where there exists a need to ensure the quality of the process and the variables in each step of fabrication is the food industry, where standards or consumer needs are a requirement of the process [4]. In recent years, the food industry has been affected by the rapid and constant changes motivated by Industry 4.0, which enables the automation and the integration of smart systems, resulting in the appearance of new challenges [5,6]. Artificial intelligence [7], big data, machine learning [8], and deep learning [9], among other technologies, provide opportunities to the industry for its growth and development, enabling a better understanding of the process to be obtained directly from its variables.

Taste recognition is one of the fields where sensors have proven their utility because allowing the industrial development of this task reduces the errors in the classification of substances [10,11,12,13]. A group of expert individuals with trained taste buds are presently working on developing this task, ensuring that the quality of the manufacturing process is certified. Although this is a good idea in the evaluation of some products, this may not be a useful solution when this process is performed on a large scale because of the number of required trained persons. However, the use of sensors enables the automation of the process by reducing possible errors and providing information for further analysis [14,15].

Taste recognition can be tackled using electronic tongues [16,17], which have provided some essential solutions in the verification of the product during the fabrication steps and in the final step prior to marketing [18]. Given the information provided by this kind of sensor, the analysis of the data is normally performed from a pattern recognition point of view [19]. The integration of these techniques, communications, and the use of new technologies in sensing and data acquisition enables this kind of system to be transferred from research to the industry, resulting in companies that can compete in the global market [20].

In order to address some of the challenges related to sensor arrays, the aim of this study was to develop a practical solution that specifically focuses on the treatment of data acquired from the sensors. Depending on the type of data collected and the sensors used, the acquisition system can generate varying data, which requires the use of effective pre-processing strategies. The aim of this study was to develop a pattern recognition methodology for sabajon classification that will help analyze and interpret these data more efficiently.

Different applications can be found for this kind of system in the literature; for instance, pharmaceutical applications [21], beer taste detection [11], trinitrotoluene (TNT) detection [22], analysis of cheeses made from raw and pasteurized cow’s milk [23], evaluation of eight kinds of flavor enhancers for umami taste [12], umami taste detection from semitendinosus sous-vide beef [24], taste evaluation of pediatric medicines [25], discrimination of honey samples based on their floral types [26], discrimination of coffee samples of different geographical origin [27], detection and classification of heavy metals [28], classification of genuine and false honey [29], and yogurt classification [13], among others [30].

As previously introduced, this work is oriented toward the classification of sabajon, which is an alcoholic beverage produced in some regions of Colombia. According to law 365 of 1994, issued by the Ministry of Health of the Republic of Colombia, in Article 6, Numeral 6.3, sabajon is defined as “a non-wine appetizer added to organic food products”. The same law also defines sabajon as “the product obtained by mixing milk, eggs, sugar, with the addition of certified neutral ethyl alcohol, brandy or other liquors and additives permitted by the Ministry of Health, with a graduation between 14 and 20 degrees of alcohol”. Because of the industry’s growth [31] and the need to ensure the quality of the product and maintain the patterns that differentiate the sabajon produced in one region from another, the use of systems such as electronic tongues is necessary to provide a reliable solution for these small companies. Including this technology can provide advantages to ensure the quality of the process and the final product or to certify the designation of origin.

The paper is organized as follows. This introduction is followed by a brief theoretical background in Section 2. The next section describes the methodology and includes the description of the steps in the methodology by describing the experimental setup and its results to show the kind of results in the classification task. Finally, the conclusions are presented in the last section.

## 2. Conceptual Framework

As a first proposal to find a solution to the problem previously introduced, this paper proposes a sabajon classification methodology that considers the use of an electronic tongue, signal processing, and pattern recognition strategies. The methodology is validated with some sabajon samples from the region of Boyaca, Colombia, showing the advantages of its use. For a better understanding of the methodology, this section presents some of the basic concepts involved in the development. These are introduced briefly because some of them are well-known in the literature. However, we suggest reviewing the corresponding references to have more details about the different concepts.

### 2.1. Electronic Tongues

An electronic tongue is a combination of several components that function in harmony to provide accurate data analysis and interpretation. These components comprise sensors, data acquisition systems, multiplexers, and a data analysis system [32]. Figure 1 shows a typical experimental setup with an electronic tongue system. First, it makes use of a sensor or a sensor network whose main objective is the interaction with the liquid or substance being analyzed [33]. There are many kinds of sensors that are normally associated with the kind of analysis and interaction with the substance to be analyzed [34]. Typically, liquids are monitored using a variety of sensors, including, but not limited to, optical, electrical, biosensors, electrochemical, and gravimetric sensors. That being said, electrochemical sensors—specifically potentiometric, amperometric, and voltammetric sensors—are the most frequently utilized sensors in electronic tongues.

Typically, when various sensors and channels are involved in capturing the signals within a data acquisition system, it is often required to implement a multiplexer system. It enables the capture of information from multiple sensors to share one device or resource.

A pre-processing step can be used if it is necessary to filter and amplify the signals from the sensors. However, these steps can be applied in an analog or a digital way, depending on their implementation. Currently, the data acquisition systems (DAQ) enable easy digital implementation of these steps. The DAQ enables the organization of the data in some files for future analysis. However, this kind of system, in some cases, includes data analysis capabilities and provides an interface to implement a different kind of algorithm. This is the case with embedded systems.

Regarding this particular work, the electronic tongue is composed of a configuration of 8 sensors which have been detailed in Table 1. Screen-printed electrodes (SPE sensors) were purchased from the company BVT Technologies, and these are made of 5 different working electrode materials. These are connected to the potentiostat through a MUX8R2 multiplexor.

### 2.2. Dimensionality Reduction

When a huge volume of data are generated, analyzing these data can be a problem because of the time and resources required [35]. One solution to this problem is the use of dimensionality reduction techniques. These techniques take advantage of the fact that not all the attributes are important for training machine learning algorithms [36]. Different techniques exist for this aim; some are principal component analysis (PCA) [37], linear discriminant analysis (LDA), projection pursuit (PP), kernel principal component analysis (KPCA), multidimensional scaling, isomap [38], non-linear PCA, singular value decomposition (SVD) [39], local linear embedding (LLE) [40], linear discriminant analysis (LDA), latent semantic analysis (LSA), locality preserving projections, and independent component analysis (ICA) [41], among others. Details about each technique and its differences can be found in [42].

### 2.3. Multivariate Data Analysis

Due to the characteristics of the electronic tongue system data, it is important to take into account the utilization of multivariate data analysis. Two main ways for the study are regression and classification methods. In the first case, these can be classified into linear and no-linear methods, and in the second case, these are organized into supervised and unsupervised methods. Some examples of supervised methods are linear LDA, PLS-DA, and KNN, among others. Examples of unsupervised algorithms are K-means, self-organizing maps, among others. In the case of this work, the methodology considers the use of classification methods.

### 2.4. Classification Methods

As we have seen before, there are various machine-learning classification methods available. In this work, we have evaluated six of these methods. Below, we have provided a brief overview of each method. As most of these methods are commonly used in the classification process, we recommend referring to the references for a more detailed understanding of their implementation and associated mathematics.

#### 2.4.1. Decision Trees

This method is one of the most popular methods for prediction and classification. This is a flowchart that is organized as a tree structure [43]. The process of splitting nodes occurs at various levels, and every node signifies a test conducted on an attribute. Meanwhile, each branch is indicative of the outcome of the test, and every leaf node (terminal node) contains a class label [44].

#### 2.4.2. Random Forest

This is a supervised method that can be used for regression and classification. In the case of classification, it uses multiple decision trees to obtain different predictions as outputs. In the process, each decision tree votes for one of the classes in the random forest, and the algorithm review which class had big voting to define it as an output [45].

#### 2.4.3. MLP Neural Network

MLP is the acronym for multi-layered perceptron; this is a neural network that uses back-propagation to separate non-linear data. It consists of different layers with nodes. Each node considers a weight, and normally, at least three layers are considered, input layer, hidden layer, and output layer [46]. In the algorithm, each node is interconnected to the nodes in the previous and after layers.

#### 2.4.4. Ada Boost

This method starts with adjusting a classifier on the dataset and continues with a sequence of classifiers working on the adjustment of the weights to correct the errors in the classification [47].

#### 2.4.5. Naive Bayes

This is a classifier that makes use of the Bayes theorem. It considers that predictive variables are independent among them; this means that a feature in the dataset is completely unrelated to the presence of any other feature [30].

#### 2.4.6. Quadratic Discriminant Analysis

This method makes use of a quadratic decision surface for the classification process. It enables the organization of two or more classes by assuming that the covariance of each class is not necessarily the same [48].

## 3. Sabajon Classification Methodology

To analyze the data from the sensors and produce accurate classification, the methodology involves several steps. These steps are carefully considered to ensure that the results obtained are reliable and can be used for decision-making. It starts with data acquisition, followed by the data scaling, after the unfolding and dimensionality reduction, and finishes with the classification of the data by using a classifier to determine the sabajon under test. An additional step is applied for the validation by using cross-validation. Figure 2 shows the steps in the methodology. The following subsections will show more details about each step and information about the experimental setup.

### 3.1. Experimental Setup

Intending to understand the steps of the methodology with its practical implementation, the following includes a description of the experiments and the kind of data that are analyzed in the methodology.

Experiments were applied to different sabajon from companies that produce sabajon in the department of Boyacá, Colombia. Among them are the Cipres and don Joaquin companies. The products were purchased directly by the authors in supermarkets in the region of Tibasosa and Sogamoso, thus seeking to test the consumable products, and the choice of the brand directly was random but guaranteed that the selected one had different varieties to validate the methodologies. Amperometric tests of multiple steps were applied to five sabajon (see Figure 3). In total, 20 experiments were developed for each sabajon, as is shown in Table 2.

In order to conduct the experiments on the five distinct sabajon variations, a combination of 40 mL of sabajon and 120 mL of de-ionized distilled water was procured. De-ionized water was obtained from a local provider in the city of Bogota, Colombia. Figure 4 and Figure 5 show the kind of signal applied and the signals acquired by the sensors. In this case, 10 different levels of 0.1 s of duration with a sample time of 0.0005 s were configured for the testing process. This means that 2000 signal points are considered per step.

### 3.2. Data Acquisition System

Data acquisition considers the use of a PalmSense4 [49] from the company PalmSens. This is a potentiostat/galvanostat/impedance analyzer that enables interaction with the sensors to acquire the information for further analysis. Because of the limited number of inputs, a multiplexer 8 to 1 is used to capture the information from all the sensors, as is shown in Figure 1.

This methodology considers the advantages of sensor data fusion because sensors are from different working electrodes which involve the integration of data from different sensors to extract information from different points of view. This means that each sensor interacts in a different manner with the liquid substance and these differences can bring elements in the combined analysis of the sensors.

### 3.3. Scaling

Because of the differences in the signals due to different sensors, scaling is a step necessary to consider in the analysis of the information from different sources in the same manner. To normalize the measurements of the sensor in the same column, the Mean Centered Group Scaling (MCGS) method is utilized. This involves taking into account the mean of all the measurements and applying it for normalization purposes [50]. More details can be found in [18]. Figure 6 and Figure 7 show the results of this step.

Figure 6 shows the platinum sensor (#2) that registers high current values compared with the rest of the sensors reaching values of 500 mA; in contrast, values in the *y* axis (Figure 7) are between −0.2 and 0.2, showing the effect of the normalization process.

### 3.4. Unfolding Data

The data to be analyzed come from different sensors, which implies that data are organized by sensor, resulting in multiple matrices. This is currently organized as a three-dimensional matrix with dimensions *experiments* × *sensors* × *samples*. The unfolding step is applied to facilitate the analysis of the data, the sensor data fusion, and the application of the rest of the steps in the methodology. It consists of the organization of the data in a bi-dimensional matrix with dimensions *experiments* × (*sensors* × *samples*). Because we use eight sensors, this matrix is organized considering 8 × 2001 = 16,008 per experiment.

According to Table 2, 100 experiments were carried out for all the sabajon samples. This means that the unfolding matrix has dimensions of 100 × 16,008. Figure 8 shows the unfolding representation by using the eight signals from the sensors for an experiment with the coffee sabajon. Figure 9 shows the outcomes of scaling the signals. However, it is noteworthy that the gold (#1) and platinum (#2) sensors exhibit a significant change in their magnitudes post-scaling. This change is quite remarkable, and it can be observed easily. Moreover, the scaling process has made the signal magnitudes from different sensors homogeneous. Therefore, none of the sensors will have a higher magnitude than the other.

### 3.5. Dimensionality Reduction

To reduce the amount of information to be analyzed and have a reduced version of this information, a dimensionality reduction algorithm is required. This step is necessary when multiple sensors are used, and the analysis of the information can be more complex because of the number of samples and the information size. In this case, seven manifold learning algorithms are tested, Hessian LLE, Isomap, Laplacian Eigenmaps, LLE, LTSA, and modified LLE. Their use enables the feature extraction process.

All these algorithms require the definition of a parameter called *k* to define the neighbors necessary to determine the neighborhood graph. For this work, it was defined in the following manner, Hessian LLE *k* = 92, Isomap *k* = 25, Laplacian Eigenmaps *k* = 20, LLE *k* = 25, LTSA *k* = 48, modified LLE *k* = 48. The results of the first three dimensions after applying the manifold learning algorithms to the sabajon dataset are evidenced in the scatter diagrams of Figure 10. In the figure, each color represents a different class to a different sabajon according to Table 2 and Figure 3. Class 1 corresponds to traditional sabajon, class 2 to brandy sabajon, class 3 to coffee sabajon, class 4 to feijoa sabajon, and class 5 to peach sabajon. The clear separation between the five classes of sabajon is evident, except for the LLE method, which places classes 2 and 4 on a line. This kind of plot enables evidence by the visual inspection of the differences between the data from each type of sabajon. However, a machine learning algorithm is required to avoid the interpretation of the plots.

### 3.6. Classification and Cross Validation

As evident in the previous results, a visual inspection in Figure 10 can be a good option to detect differences between all the classes; however, from our experience, there exist datasets where data are not separated, and visual inspection can not be an option. The following is a description of the elements considered in the configuration of the classifiers.

It is necessary to determine the target dimension to use the Laplacian Eigenmaps algorithm. To identify this parameter, a study was conducted on the relationship between the target dimension, denoted as *d*, and the accuracy of the MLPNN algorithm in classifying the data. A leave-one-out cross-validation process (LOOCV) was used to carry out this analysis because of the limited number of samples in the sabajon dataset. Accuracy behavior tends to increase as *d* increases up to a maximum of *d* = 5 for a classification accuracy of 100%, as shown in Figure 11. The 100% accuracy confusion matrix is shown in Figure 12.

Table 3 shows the accuracy values for the final comparison of classifiers against dimensionality reduction methods. In general terms, high values of classification accuracies were obtained, most of them reached values of 98%. The worst classifier was Adaboost because when it was used to classify the data reduced with LTSA it obtained an accuracy classification value of 38%. In addition, when it classifies the data reduced with the algorithm Isomap, the classification accuracy was 59%. On the other hand, the best classifier was the MLPNN because it obtains a classification accuracy of 100% in all cases; in the same way, the best dimensionality reduction method was the Laplacian Eigenmaps because a classification accuracy of 100% was obtained in all cases for every classifier. These differences in the results show that different machine learning produces different results in the methodology. Results show that one of the main factors is the type of classifier and the data reduction method combination. Although all the algorithms are oriented to classify the kind of data and the parameters of the algorithm, how it is trained enables us to obtain different results. Considering these factors is important when selecting and implementing machine learning methods to achieve the best possible results.

## 4. Conclusions

We recommend utilizing Laplacian Eigenmaps as the manifold learning algorithm and MLP as the classifier to achieve optimal results. It is essential to calibrate the parameters of these algorithms for optimal classification accuracy, with the neighborhood graph construction parameter (*k*) and the number of target dimensions being critical factors. Normalizing data is also crucial in analysis, enabling uniform data comparison across different sensors. Overall, our methodology represents a new contribution to the classification of liquor beverages and has significant potential for further research and development.

In future work, it would be beneficial to conduct additional experiments with diverse sabajon varieties sourced from various regions of Colombia. This will help determine whether there are any variations in flavor profiles despite using the same ingredients from different producers. This kind of experiment will enable the analysis of the methodology by considering the product’s origin to determine possible differences. In addition, it is necessary to evaluate the methodology with another kind of alcoholic beverage substance to determine its applicability in the classification of different beverages. This methodology can be extended as a hypothesis to classify different liquid substances because it only requires data captured in the same manner with this kind of sensor. In that case, the methodology result will classify the newly analyzed substances. A comparison of latency and facilities for their implementation in an embedded system also requires evaluation for future deployments in portable systems.

## Figures and Tables

**Figure 1 sensors-23-06178-f001:**
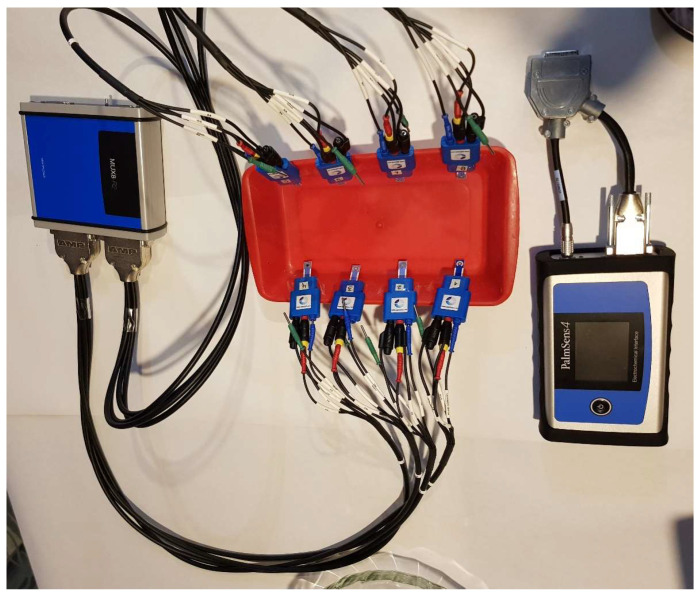
Electronic tongue setup.

**Figure 2 sensors-23-06178-f002:**
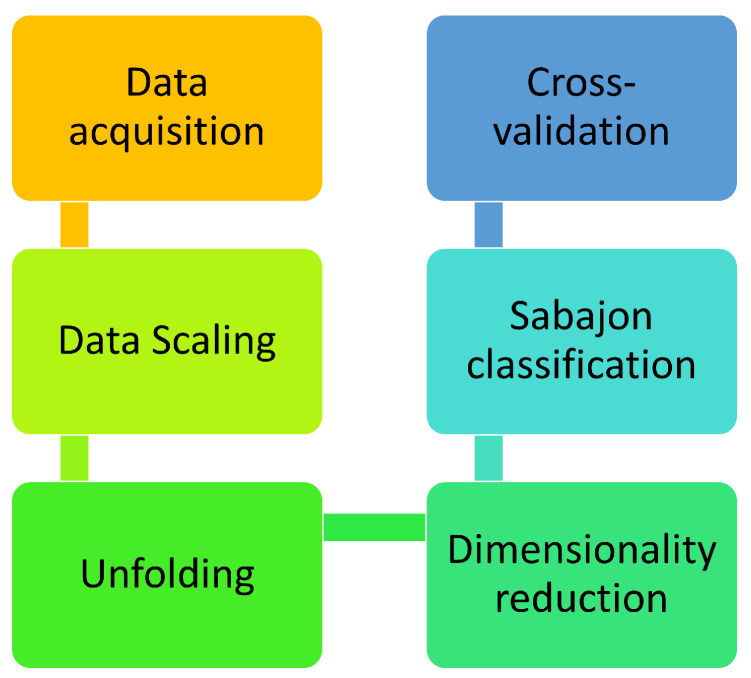
Sabajon classification methodology.

**Figure 3 sensors-23-06178-f003:**
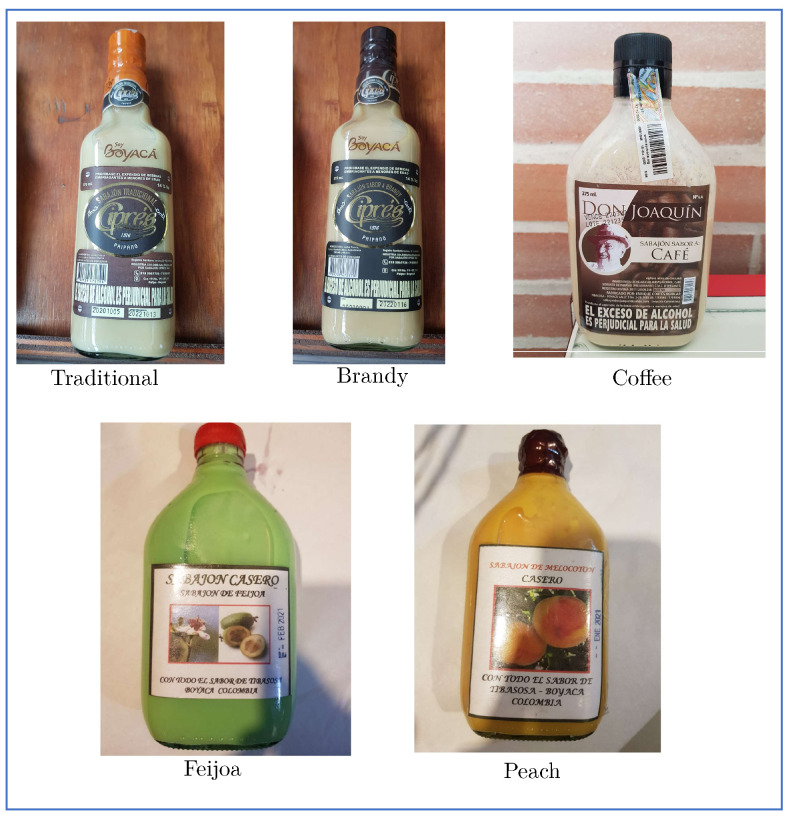
Photo of the different Sabajon evaluated.

**Figure 4 sensors-23-06178-f004:**
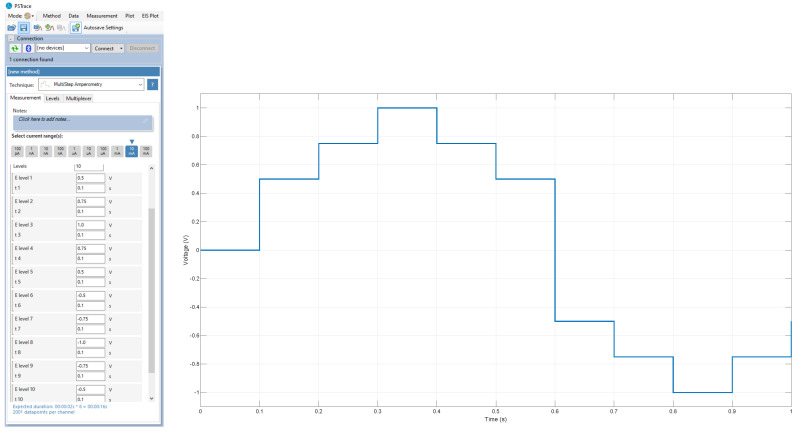
Signal applied to the sensors. The figure shows current vs. time.

**Figure 5 sensors-23-06178-f005:**
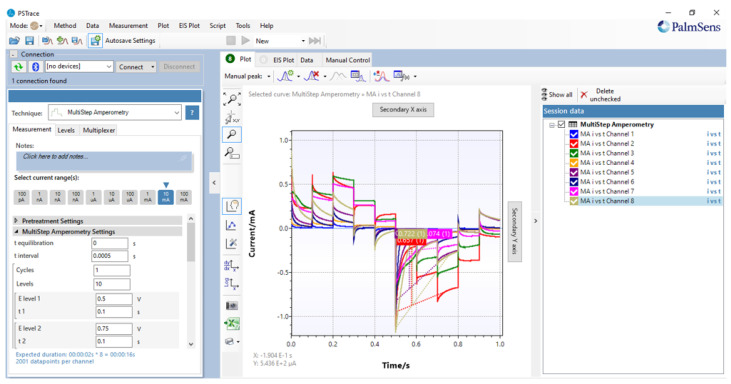
Signals obtained by the sensors by using multi-step amperometry and visualized by PSTrace interface.

**Figure 6 sensors-23-06178-f006:**
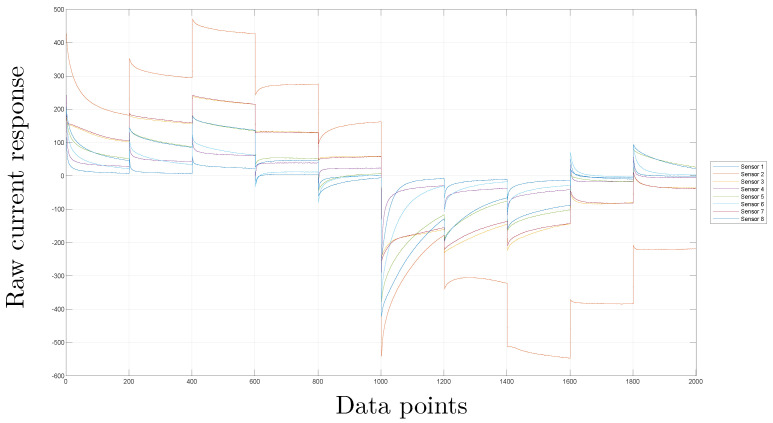
Signals from the eight sensors without pre-processing for one of the experiments with coffee sabajon.

**Figure 7 sensors-23-06178-f007:**
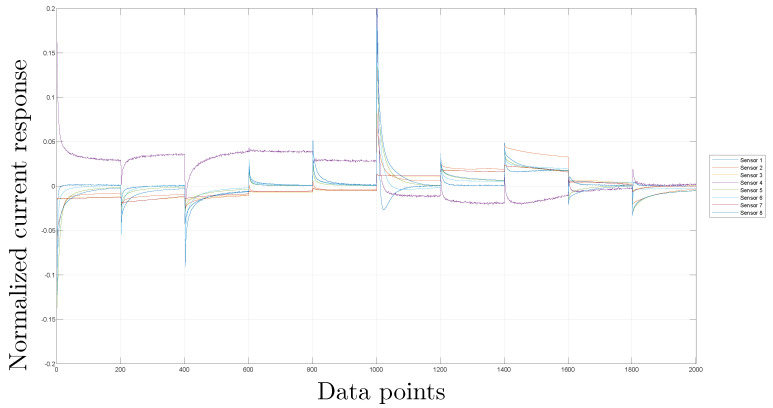
Signals from the eight sensors with the scaling step for one of the experiments with coffee sabajon.

**Figure 8 sensors-23-06178-f008:**
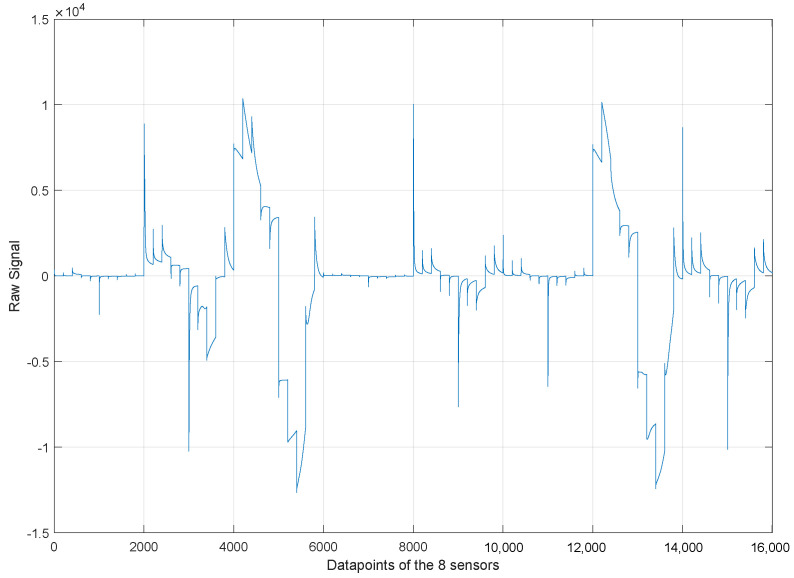
Unfolded matrix with the 8 sensors without scaling for one experiment with coffee sabajon.

**Figure 9 sensors-23-06178-f009:**
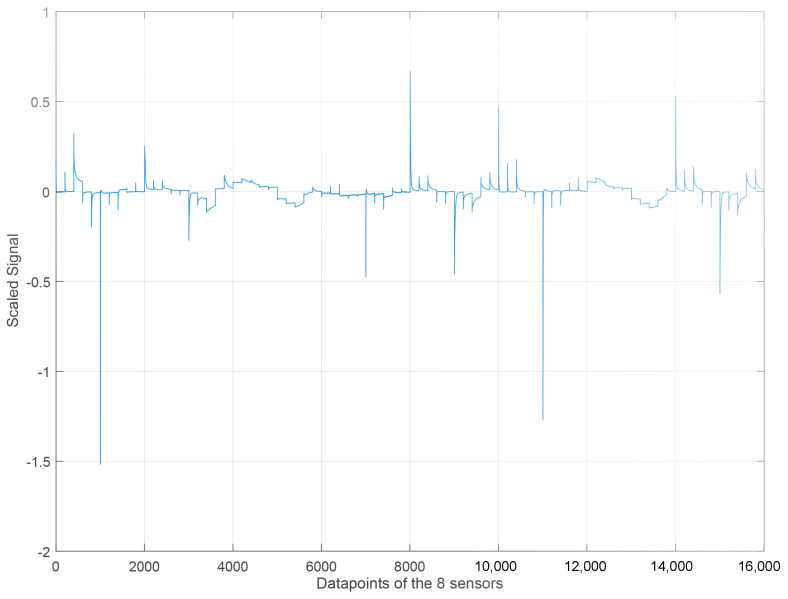
Unfolded matrix with the 8 sensors with the scaling step for one experiment with coffee sabajon.

**Figure 10 sensors-23-06178-f010:**
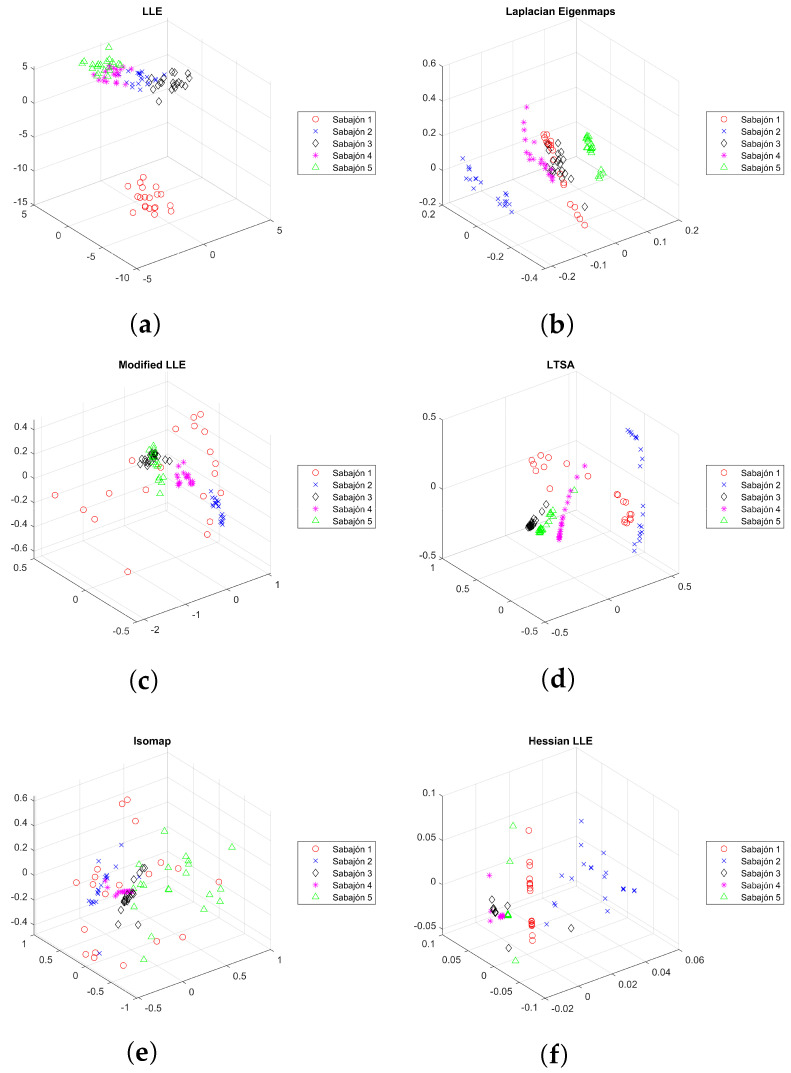
Scatter diagrams of the first three dimensions resulting from applying the dimensionality reduction methods on the sabajon dataset. (**a**) LLE, (**b**) Laplacian Eigenmaps, (**c**) Modified LLE, (**d**) LTSA, (**e**) Isomap, and (**f**) Hessian LLE.

**Figure 11 sensors-23-06178-f011:**
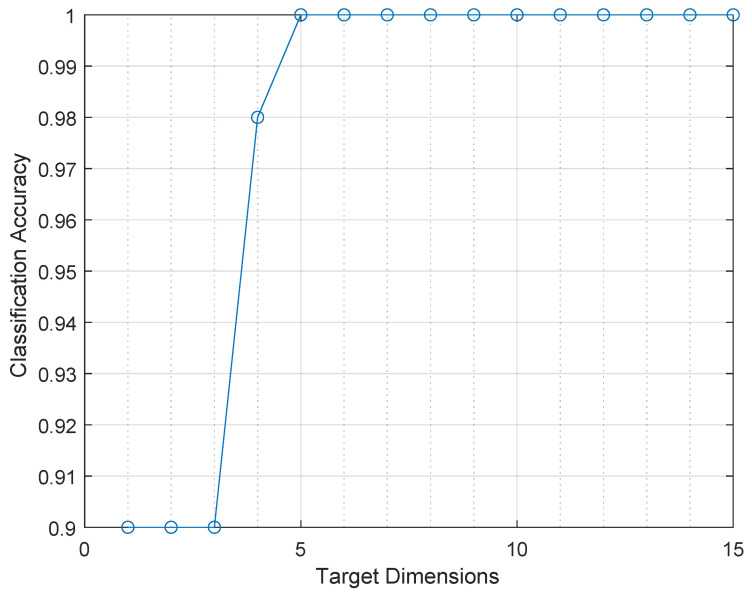
Accuracy behavior when varying the number of dimensions obtained with the Laplacian eigenmaps method at the input of the MLPNN classifier algorithm.

**Figure 12 sensors-23-06178-f012:**
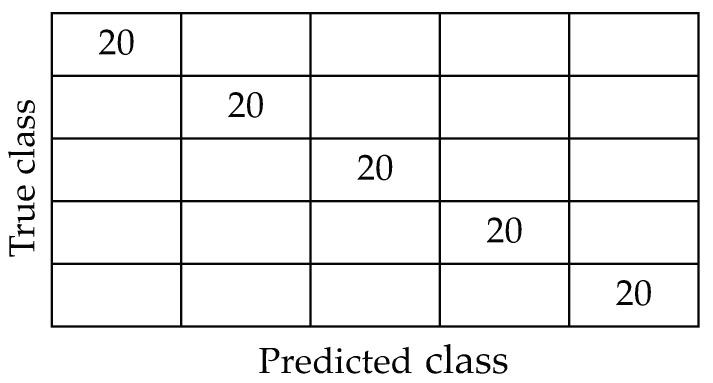
Confusion matrix resulting from using Laplacian Eigenmaps and multi-layer perceptron neural network with the LOOCV in the sabajon dataset. Taking into account eight dimensions at the input of the MLPNN algorithm.

**Table 1 sensors-23-06178-t001:** Sensors in the electronic tongue.

ID	Working Electrode Material
S1	Gold
S2	Platinum
S3	Silver
S4	Graphite
S5	Platinum
S6	Gold-Platinum Alloy
S7	Silver
S8	Platinum

**Table 2 sensors-23-06178-t002:** Information about the kind of sabajon tested.

ID	Sabajon	Number of Experiments
1	Traditional	20
2	Brandy	20
3	Coffee	20
4	Feijoa	20
5	Peach	20

**Table 3 sensors-23-06178-t003:** Classification accuracy behavior changing the classifiers versus dimensionality reduction methods.

	Decision Tree	Random Forest	MLP Neural Network	AdaBoost	Naive Bayes	Quadratic Discriminant Analysis
LTSA	0.98	0.98	1	0.38	1	1
LLE	0.99	0.99	1	0.99	0.99	1
Hessian LLE	0.98	1	1	0.99	0.97	1
Modified LLE	1	0.99	1	0.99	1	1
Isomap	1	0.99	1	0.59	0.98	0.98
Laplacian Eigenmaps	1	1	1	1	1	1

## Data Availability

The data presented in this study are available on request from the corresponding author. The data are not publicly available due to some doctoral theses are working and increasing the information in the dataset.

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
