# Peer review of "New Electronic Tongue Sensor Array System for Accurate Liquor Beverage Classification"

_sensors, 2023, doi:10.3390/s23136178_

Round 1

Reviewer 1 Report

Taste recognition is one of the fields where sensors have proven their utility in the food industries due to error reduction and accurate classification of substances. The study explores applying an electronic tongue system in categorizing and validating the methodology of sabajon, an alcoholic beverage produced in some regions of Colombia per the law 365 of 1994, issued by the Ministry of Health of the Republic of Colombia, in article 6, numeral 6.3. The research result merits the work to be considered for publication because the finding shows high accuracy in all samples tested per the utilized signal processing techniques from a series of tongue-type electronic sensors.

However, the following observations are recommended for consideration to merit the work for publication.

·      Line 31: “challenges[5], [6].” Check space

·      Line 36: change “Taste Recognition” to “Taste recognition”

·      Line 36-38: “Taste Recognition is one of the fields where sensors have proven their utility because allowing the development of this task in an industrial way by reducing the errors in the classification of substances” statement isn’t clear consider this “Taste Recognition is one of the fields where sensors have proven their utility because allowing the development of this task in an industrial way reduces the errors in the classification of substances”

·      Line 40-42: “Although this is a good idea in the evaluation of some products, this is not a useful solution when this process is performed on a high scale because the number of required trained persons” statement isn’t clear consider this “Although this is a good idea in the evaluation of some products, yet this could not be a useful solution when this process is performed on a high scale because of the number of required trained persons”

·      Line 62: define TNT

·      Line 63: change “Evaluation” to “evaluation”

·      Line 81-84: move the statement to section 2 “conceptual framework” to make the write-up well organized and scientific.

·      Line 108: “when there are various sensors and channels involved in capturing” statement isn’t clear consider this “when various sensors and channels are involved in capturing”

·      Line 122: define SPE

·      Line 123: define BVT

·      Line 142: define PLS-DA

·      Line 142: change “Linear” to “linear”

·      Line 143: define KNN

·      Line 143: change “Unsupervised” to “unsupervised” and “Self” to “self”

·      Line 148:  rewrite “following there is some” as “following a”

·      Line 162: correct “review”

·      Line 163: define MLP

·      Line 187-188: delete statement

·      Line 197: give the names and locations of the supermarket Sabajon products that were secured for analysis

·      Figure 3, 6,7: there’s a period after the caption check and delete as in Figure 2, or if needed apply to other figure captions areas where necessary.

·      Line 203: give the location or source where the 120 ml of de-ionized distilled water was procured.

·      Line 216: change “Because of the differences in the signals because of the different sensors” to “Because of the differences in the signals due to different sensors”

·      Line 230: change “is applied” to “are applied”

·      Line 231: change “consists on” to “consists of”

·      Section 3.5: correct “3.5. Dimmensionality reduction” to “3.5. Dimensionality reduction”

·      Line 272: change “100%.” To “100%”

·      Line 276-278: revise for clarity

·      Line 278: “38 %” fix space as “38%”

·      Line 278-279: give a clear interpretation of the 59% accuracy inferred

·      Line 316: revise for clarity “despite…..”

·      Line 317: “This analysis will allow the analysis by considering the origin of the product” Rewrite the statement for clarity

·      Line 320: revise for clarity

·      The abstract section indicates “This paper explores the application of an electronic tongue system in categorizing sabajon, an alcoholic beverage found in specific regions of Colombia” but the study didn’t reveal the exact sabajon categorization in sections 3 and 4. Authors are recommended to revise the section where necessary for clarity.

· T methodology, the experimental setup, and results in the classification task synergy aren’t well revealed in the study. Authors are recommended to revise the section 3 results by discussing the study classifications outcome where necessary for clarity and as well cite existing literature for an improved paper.

·      Which sensor was the best electrode working material from the study?

·      State the best working sensor material in the abstract section.

·      Among the 5 sabajon products which product brand has the best e-tongue sensor detection and processing accuracy

Reviewer 2 Report

Reviewers' comments:

This article (sensors-2448280) is entitled “New Electronic Tongue Sensor Array System for Accurate Sabajon Classification as a Liquor Beverage” by research group Diego A Tibaduiz. This article is about sensor array system for accurate sabajon cassification.

(A) RECOMMENDATION

* Minor revision

 (B) COMMENTS TO AUTHOR

TITLE:

Title: why only for sabajon, is this sensor applicable for other alcoholic beverage

ABSTRACT:

Overall abstract is difficult to follow, terms like classification methodology, categorizing sabajon are confusing; revise the abstract with the simplicity of the method and clarity of the objective.

INTRODUCTION:

The introduction contains several trivial statements ad descriptive statements; readers will be bored when reading such statements. The introduction should focus on technical challenges in sensor technology. Improve connectivity and flow  .  

MATERIALS AND METHODS:

A subsection of 2.4. Classification methods, somehow it is difficult to understand and minute details are needed to reproduce protocols.

RESULTS:

Figure 3. Different Sabajon evaluated. The question is is this sensor technology only applicable to Sabajon

Subsection 3.2. data acquisition system: provide minute details.

3.6 what is the importance of cross validation

 5. Conclusion

Focus on novelty in method, provide conclusions in scope of sustainability and cost-effectiveness, comment on future implications

Minor revision

Round 2

Reviewer 1 Report

Authors have responded to the comments critically and have improve the quality of the paper after a thorough revision. The manuscript is not only revised but it is complemented with detailed scientific justifications at sections where necessary. Hence the manuscript merit to be accepted for publication.

Author Response

Thank you for all the comments and suggestions. All of them allow us to improve the quality of our paper.

Reviewer 2 Report

The authors have sufficiently revised the manuscript and answered the reviewers' comments.  Revise the conclusion section: make it short,  straightforward, and precise. 

Minor check 

Author Response

According to the suggestion, the conclusion section was reviewed and changed in the document.